# Humoral and Cellular Immune Response after Three Doses of Sinopharm [Vero Cell]-Inactivated COVID-19 Vaccine in Combination with SARS-CoV-2 Infection Leads to Hybrid Immunity

**DOI:** 10.3390/ph17010122

**Published:** 2024-01-17

**Authors:** Marija Vukčević, Katarina Šerović, Mateja Despot, Aleksandra Nikolić-Kokić, Aleksandra Vujović, Milan Nikolić, Duško Blagojević, Tanja Jovanović, Dragana Despot

**Affiliations:** 1Institute for Biocides and Medical Ecology, Trebevićka 16, 11030 Belgrade, Serbia; marija.vukcevic@biocidi.org.rs (M.V.); katarina.serovic@biocidi.org.rs (K.Š.); zavod@biocidi.org.rs (D.D.); 2Faculty of Medicine, University of Belgrade, dr Subotića starijeg 8, 11000 Belgrade, Serbia; drdespotmateja@gmail.com; 3Institute for Biological Research “Siniša Stanković”-National Institute of Republic of Serbia, Department of Physiology, University of Belgrade, Despota Stefana 142, 11108 Belgrade, Serbia; san@ibiss.bg.ac.rs (A.N.-K.); dblagoje@ibiss.bg.ac.rs (D.B.); 4Dr “Simo Milošević“ Health Center, Požeška 82, 11030 Belgrade, Serbia; alexandra.vujovic5@gmail.com; 5University of Belgrade, Faculty of Chemistry, Department of Biochemistry, Studentski trg 12-16, 11158 Belgrade, Serbia; mnikolic.chem@gmail.com

**Keywords:** COVID-19, Sinopharm vaccine, neutralizing antibodies, anti-S1 IgG antibodies, anti-RBD IgM antibodies, anti-N IgM antibodies, anti-N IgG antibodies, INF γ

## Abstract

Background: Several vaccines against COVID-19 have been developed and licensed to enhance the immune response against SARS-CoV-2. Similarly, previous infection with SARS-CoV-2 has been shown to provide significant protection against severe infection and hospitalization. Methods: We investigated the effect of three doses of the Sinopharm vaccine and SARS-CoV-2 infection on the specific immune response in 103 volunteers, measuring neutralizing antibodies, anti-S1 IgG, anti-RBD IgM, anti-N IgM, anti-N IgG antibodies, and INF γ. Results: Our results showed that the presence of cardiovascular diseases increased the level of anti-N-IgG antibodies, while endocrinological diseases decreased the level of neutralizing antibodies and anti-N IgG antibodies, suggesting that these diseases alter the effect of vaccine-induced immunity. In addition, there was a significant decrease in anti-S1 IgG levels at 6 months and in anti-N IgG levels 18 months post-infection, while neutralizing antibodies and INF γ levels were constant at 3, 6, and 18 months post-infection. Conclusions: Our results confirm the emergence of hybrid immunity, which is the strongest and most durable compared to natural immunity or vaccine-induced immunity. Significant positive correlations were found between humoral and cellular immunity markers: neutralizing antibodies, anti-S1 IgG and anti-N IgG antibodies, and INF γ, indicating a unique coordinated response specific to COVID-19.

## 1. Introduction

SARS-CoV-2 belongs to the family Coronaviridae, genus Betacoronavirus. The morphology and structural features of the new virus are identical to those of other human coronaviruses. Although the mutation rate was low in the beginning, the number of variants, some of which are of concern (alpha, beta, gamma, delta, omicron variants, and subvariants of omicron), has increased with the prolonged spread of the virus [1]. Previously acquired immunity, induced through infection or vaccination, leads to cross-protection against severe clinical forms of the disease, even against infections caused by the new variants of the virus [2,3,4,5]. The quantities of SARS-CoV-2 antibodies vary widely among patients and depend on numerous factors, such as the severity of clinical presentation, age, associated comorbidities, and patients’ immunocompetence, as well as the methods used to measure specific antibody titers [5,6]. Patients with certain diseases, such as cardiovascular (CVD) [7] and endocrinological diseases [8], are particularly vulnerable to COVID-19, and these patients are more prone to developing severe clinical presentations of the infection, such as critical conditions and death. Therefore, such individuals were of particular interest to our research. However, despite the differences in serological values, reinfections by the same viral variant were not as frequent. This is confirmed by numerous studies conducted after the introduction of the vaccine, which showed that seropositive individuals had a significantly lower risk of reinfection than seronegative individuals at least six months after the previous infection [9]. Numerous studies have demonstrated differences in the quality of the immune response following infection/vaccination based on the tracking of markers of humoral and cellular immunity [10]. Although there are no specific correlates for the protective role of the immune response, the most important marker of the efficiency of the immune response is the presence of neutralizing antibodies, namely anti-RBD epitope antibodies and anti-S1 antigen antibodies, which enable the virus to bind to the ACE2 receptor [9]. Given the importance of the cellular immune response, especially CD4+ and CD8+ T lymphocytes, for the efficient elimination of infected cells, the most common marker of the protection provided by cellular immunity is the concentration of interferon-gamma [11,12]. Most of the licensed vaccines induce immunity against the S protein. These vaccines are mainly based on mRNA and recombinant DNA technology and are designed to induce vaccine immunity against the viral S antigen; therefore, they induce no change in nucleocapsid antibody titers in immunized individuals [13,14,15]. The difference between these vaccines and the Sinopharm vaccine is that the latter contains completely inactivated virus particles. Therefore, following the immune response after immunization with the Sinopharm vaccine is particularly interesting, as it is expected to induce immunity against all viral antigens, including nucleocapsid antigens. Also of interest was the investigation of hybrid immunity, i.e., the immune protection of individuals vaccinated with the COVID-19 vaccine in whom at least one SARS-CoV-2 infection was registered before or after vaccination. In other words, we investigated the influence of the vaccine on naturally acquired immunity and *vice versa*.

## 2. Results

Humoral and cellular immune responses in 103 individuals (28 males and 75 females) after SARS-CoV-2 infection and six months after receiving the third dose of the Sinopharm vaccine showed positive values for anti-SARS-CoV-2 neutralizing antibodies, anti-N IgG antibodies, anti-S1 IgG antibodies, and the cellular immune response (IFN γ), while levels of anti-RBD IgM antibodies and anti-N IgM antibodies were considered negative, as the values were lower than <18 U/mL and <0.8.

Our results showed that age (participants were divided into ten-year interval age groups: 20–30; 30–40; 40–50; 50–60) did not influence the analyzed parameters: humoral anti-SARS-CoV-2 neutralizing antibodies-NA (F = 1.02, *p* > 0.05), anti-S1 IgG antibodies (F = 0.21, *p* > 0.05), anti-N IgG antibodies (F = 0.17, *p* > 0.05), and cellular immune response (IFN γ) (F = 1.03, *p* > 0.05). When we divided all the patients by sex, the results similarly showed no difference between the analyzed parameters: humoral anti-SARS-CoV-2 neutralizing antibodies-NA (F = 12.58, *p* > 0.05), anti-N IgG antibodies (F = 1.029, *p* > 0.05), anti-S1 IgG antibodies (F = 0.492, *p* > 0.05), and cellular immune response (IFN γ) (F = 1.157, *p* > 0.05). However, levels of neutralizing antibodies, anti-S1 IgG, anti-N IgG, and IFN γ were significantly higher in individuals who had COVID-19 before vaccination than in those vaccinated but without a history of COVID-19 infection (Figure 1). The values of anti-RBD IgM antibodies in individuals vaccinated following a COVID-19 infection are statistically significant and enter positive values (>18 U/mL).

When all subjects were additionally separated by sex and history of COVID-19, and a two-way ANOVA was performed, results showed that sex did not affect the immune response. However, in participants who have had COVID-19, we saw increased levels of INF γ, anti-N IgG, and anti-S1 IgG antibodies (two-way ANOVA, statistically significant effect of infection) (Figure 2).

Some of the participants included in our study suffered from cardiovascular diseases (CVDs). So, when we divided the participants into those with and those without CVD, statistical analysis showed that participants with CVD produced elevated anti-N IgG levels (Figure 3).

Although IgM values were below the positivity threshold, in subjects with CVD, we saw higher anti-N IgM antibody levels to a statistically significant extent (Figure 3). Interestingly, after adding the history of COVID-19 as a second parameter and performing a two-way ANOVA, participants with CVD showed a statistically significant increase only in anti-N IgM levels regardless of whether having had the infection or not (Figure 4). Our study showed that endocrinological disorders did not influence the observed parameters: humoral anti-SARS-CoV-2 neutralizing antibodies-NA (F = 2.91, *p* > 0.05), anti-S1 IgG antibodies (F = 1.33, *p* > 0.05), anti-N IgG antibodies (F = 1.095, *p* > 0.05), and cellular immune response (IFN γ) (F = 2.069, *p* > 0.05). However, when COVID-19 infection was added as a second parameter, results showed that endocrine disorders attenuated the elevation of neutralizing antibodies in vaccinated individuals with no history of COVID-19 (Figure 5).

Our results showed that COVID-19 infection elevated neutralizing antibodies, anti-S1 IgG, and anti-N IgG antibodies (Figure 5). Since the presence of endocrinological disorders and a prior history of COVID-19 showed no significance at the level of either humoral or cellular response, except neutralizing antibodies, we singled out the group of subjects who have not had COVID-19 and divided them into two groups—with and without endocrine disorders. A statistical T-test showed that neutralizing antibodies and anti-N IgG antibodies decreased significantly in the Endo/No group, implying that subjects with endocrine disorders who were vaccinated with the Sinopharm vaccine and had no history of COVID-19 had a lower humoral response than the vaccinated subjects without endocrine disorders (Figure 6).

To determine the extent of change in antibody levels over time, participants were divided into three subgroups based on the period elapsed between the last clinical presentation of COVID-19 symptoms and the time of testing (up to 3 months, 3 to 6 months, and up to 18 months since the previous symptoms). The results show no decrease in neutralizing antibodies and IFN γ 3, 6, and 18 months after COVID-19, while there is a significant decrease in anti-S1 IgG 6 months after and anti-N IgG 18 months after COVID-19. The significance was also confirmed for anti-N IgM levels, although the value was below the threshold (Figure 7).

Correlation analysis performed on the complete sample showed that, in vaccinated individuals, significant positive correlations exist between humoral and cellular markers of immunity: neutralizing antibodies, anti-S1 IgG, anti-N IgG, and INF γ, suggesting a unique coordinated response specific for COVID-19 (Table 1).

## 3. Discussion

Vaccination against COVID-19 began in Serbia in January 2021. This study analyzed the markers of vaccine-induced immune response six months after receiving the third dose of the Sinopharm vaccine. There are several reasons why this study included only 103 subjects, but two reasons have emerged as the most important. Firstly, in our country, individuals were allowed to choose the type of booster vaccine. For this reason, in many cases, people who had received two doses of the inactivated vaccine chose the mRNA vaccine for a booster, as they did not have detectable levels of neutralizing antibodies in the days before the booster dose (unpublished data). Secondly, many people chose not to receive a booster dose of the vaccine, either because they believed they were not in the high-risk group for COVID-19 or because they already had the vaccine and did not believe they would benefit from a booster dose. For this reason, they were also excluded from the study. All participants were vaccinated with the Sinopharm vaccine with the inactivated Wuhan virus strain, regardless of whether new variants of concerns were actually circulating. Based on genome sequencing analysis of viruses isolated from nasopharyngeal swabs from March 2020 to March 2022, three epidemiological waves of COVID-19 were observed during the vaccination period in Serbia: the dominance of the alpha variant from late December 2020 to April 2021, the delta variant from July 2021 to 31 December 2021, and the Omicron variant from 31 December 2021 to March 2022 (our unpublished data). All subjects who showed symptoms of COVID-19 after infection (27 subjects) became ill from January to March 2022, when the Omicron variant prevailed. A total of 16 study participants had been infected with SARS-CoV-2 before the start of immunization from November 2020 to September 2021.

The protective immune response to SARS-CoV-2 results from a combination of humoral and cellular immune responses [16]. Of the four structural proteins of coronaviruses (S, E, M, N), the two most abundant are the S and N antigens. The surface spike S glycoprotein is crucial for the first step of infection, as it mediates viral entry by binding to the host ACE2 receptor and fusing the virus-host membrane. The S antigen consists of the subunits S1 and S2. The epitopes for neutralizing antibodies and the epitopes for cellular immune response are located within the RBD domain [17,18]. The N protein is immunodominant and is highly expressed in infected cells. Within the N antigen are epitopes for the cellular immune response of CD4- and CD8-positive T lymphocytes, as well as epitopes targeted by non-neutralizing antibodies [19]. Therefore, most tests measuring the efficacy of the immune response are based on these two antigens. It is known that antibodies produced during infection with different human coronaviruses (HcoV-OC43, HcoV-HKU1, HcoV-229E, HcoV-NL63), especially from the same genus, have a potential for cross-reactivity [20]. Antibodies against the S1 and RBD domains of the S protein were found to be hypervariable and subtype-specific. This is in contrast to the N antigen and the S2 subunit of the S antigen, which are highly conserved in human coronaviruses. However, antibodies against N and S2 antigens formed during previous infections with human coronaviruses cannot protect against SARS-CoV-2 infection [6,20,21].

Since most vaccines are designed to elicit an immune response to the S antigen only, tests that detect the presence of neutralizing antibodies are most commonly used to monitor the efficacy of vaccination [22,23,24]. However, the Sinopharm vaccine elicits a response to all individual antigens (including the S and N antigens). Hence, determining the serological profile in infected and vaccinated individuals is of great interest. According to the literature, the efficacy of inactivated COVID-19 vaccines after the administration of two doses is 70–80% [25,26]. Therefore, a third booster dose was recommended, which, according to our results, induced longer-term immunity against the original virus variant [27]. Using biomarkers of humoral and cellular immunity as indicators, we found that all recipients of three doses of the Sinopharm vaccine developed an effective humoral and cellular response regardless of gender and age.

Humoral and cellular immunity biomarkers were analyzed separately in vaccinated individuals with COVID-19. These individuals have hybrid immunity consisting of natural and vaccine-induced immunity. In the literature, this type of “hybrid immunity” is called superior immunity. In vaccinated individuals without SARS-CoV-2 infection, a significant decrease in anti-S1 IgG and IFN α levels was observed within 3 to 6 months. However, individuals who acquire natural immunity through infection and are subsequently vaccinated develop stronger immunity to SARS-CoV-2 [26,28]. Many authors emphasize the importance of memory immunity. It has been shown that the number of memory B cells is increased 5- to 10-fold in hybrid immunity compared to natural infection or vaccination alone [29]. According to the data available in the literature, previous infection alone and previous infection in combination with the last vaccination provide high and sustained protection against hospitalization or severe disease [30]. As there is limited data in the scientific literature on serological status after vaccination with the Sinopharm vaccine, we considered our results in light of serological markers detected after vaccination with an mRNA vaccine. After one dose of an mRNA vaccine, humoral immunity is 10–45 times higher in people who have already had an infection than in people who have not yet been infected [31,32]. The administration of the second dose leads to a threefold increase in antibodies in non-immune individuals but does not have this effect in individuals with a previous infection. Individuals who have been fully immunized with two doses of an mRNA vaccine and have a previous infection have six times higher antibody levels than those who have only been naturally infected or fully immunized. It has also been shown that vaccination of previously infected individuals leads to significantly higher levels of cross-neutralizing antibodies than in fully vaccinated individuals [33].

In our study, the participants who had COVID-19 before vaccination and were fully immunized with three doses of the Sinopharm vaccine showed a statistically significant increase in all types of SARS-CoV-2-specific antibodies, especially anti-S1 IgG and anti-N IgG. The titer of neutralizing antibodies was also higher in the participants with the previous infection group, as was the IFN level. Our results confirm the emergence of hybrid immunity, which provides the greatest and most durable protection based on humoral biomarkers. Several participants, especially those with COVID-19 immediately before and after vaccination, had positive anti-RBD IgM antibodies. The presence of subtype-specific IgM antibodies is likely because the virus variant in the vaccine differs from the variant that caused the infection, which stimulates the de novo production of IgM antibodies.

In addition, this study was focused on patients with diseases that may affect the immune system. This applies also to responses to vaccination, including anti-SARS-2 immunization. It is well-known that individuals with cardiovascular and endocrinological disease have a higher risk of complications and death after infection with SARS-CoV-2, and the humoral immune response is also lower in these groups of patients after natural infection [34]. It was expected that vaccination programs would increase the protection of these patients; therefore, booster vaccinations were proposed to enhance the response to viral infection. According to the literature, cardiovascular diseases, especially hypertension, is an important cofactor for severe COVID-19 [35,36,37,38]. About 85% (27 out of 32) of the patients with CVD included in our study use ACE inhibitors as part of their treatment. Studies have provided evidence that hypertension, among other cardiovascular diseases, is a risk factor for a reduced antibody response after vaccination with the mRNA vaccine; treatment against hypertension has, in contrast, been linked to clinical benefits in COVID-19. These benefits are supposedly related to improved antibody production through positive effects on inflammatory pathways and antigen presentation [34]. Given the fact that older persons in this study were treated for hypertension or other cardiovascular diseases, the positive relation between hypertension or cardiovascular disease and the antibody response could be a treatment effect. However, in our study, there were no statistically significant differences in markers of humoral and cellular immune response between patients with cardiovascular disease and other participants, except for anti-N-IgM, which was higher in CVD patients (Figure 4).

Endocrine disorders also influence the efficacy of the immune response [39,40]. According to our study, patients with endocrine disorders show a statistically significant impairment of the humoral immune response after vaccination, with lower values for all antibodies tested (neutralizing antibodies, anti-S1-IgG, anti-N-IgG), which means that subjects with endocrine disorders have a lower humoral response after Sinopharm vaccination. However, these differences were lost in the group of participants with a history of COVID-19 and endocrine disorders, and there was no difference in humoral and cellular markers.

The level of neutralizing and anti-IgG antibodies specific for each antigen decreased significantly after 3 to 6 months following two doses of the Sinopharm vaccine. An additional booster dose is essential to increase protection against COVID-19. Overall, these results show that people with a history of COVID-19 who have been vaccinated with three doses of the inactivated vaccine develop hybrid immunity. This means that participants with hybrid immunity have the best adoptive immunity, likely against both the original strain and the variants in question. Neutralizing antibody and IFN γ titers remained unchanged during the study period in our group of participants with COVID-19 infection. However, anti-S1 IgG and anti-N IgG antibodies showed a statistically significant decrease in titer when sampled 18 months post-infection (Figure 7). Our results confirm that the inactivated Sinopharm vaccine, like mRNA and vector-based vaccines, induces effective production of neutralizing antibodies. Natural immunity and vaccine-induced immunity against SARS-CoV-2 are clearly involved in protection against reinfection with SARS-CoV-2. The results also point to the contribution of the T-cell response to protection, in particular, immunological memory as a source of protective immunity. Correlation analysis also showed that immunity was established through positive cooperativity between anti-S1 IgG and anti-N IgG, which stimulate each other as well as NA and IFN γ, via independent and separate pathways (Figure 8). According to our results, this connection represents the main pathway for establishing protection in COVID-19 hybrid immunity. Therefore, our study supports the importance of vaccination for boosting immunity and suggests the possibility of a long-term antiviral immune response after administration of the Sinopharm vaccine.

## 4. Materials and Methods

### 4.1. Study Design

This study aimed to measure humoral and cellular immune response in immunized individuals six months after receiving the third dose of the SARS-CoV-2 Sinopharm vaccine. All participants included in the study voluntarily signed an informed consent form, completed a self-questionnaire (Appendix A), and had their blood drawn to perform the testing. The study was approved by the Institutional Ethics Committee of the Institute for Biocides and Medical Ecology, Belgrade (protocol number 05-01 468/3-1, approved on 23 February 2022).

### 4.2. Participant Selection and Serum Collection

All participants who met the inclusion criteria and signed the informed consent were included in the study and divided into groups based on gender, history of SARS-CoV-2 natural infection, and presence of cardiovascular, pulmonary, and autoimmune diseases, diseases of the endocrine and nervous system, and liver and kidney diseases.

Serum samples were obtained by collecting 4 to 6 mL of whole blood in VACUETTE^®^ Serum Tubes (Greiner Bio-One GmbH, Kremsmünster, Austria). The blood was centrifuged at 3000 rpm for 10 min (Gyrozen 416 centrifuge) before aliquoting the serum. Upon testing, the serum was stored at −20 °C. The samples for measuring SARS-CoV-2 T-cell specific response were obtained by collecting 4 to 6 mL of whole blood in VACUETTE^®^ Heparin Tubes (Greiner Bio-One GmbH) and processed immediately according to the manufacturer’s instructions. All the samples were collected between March and June 2022.

The study involved 103 participants in total, 28 males and 75 females. Anamnestic data for participants are presented in Table 2. Of the 103 subjects, 36 had cardiovascular diseases, 11 had endocrinological diseases, 13 had allergic reactions, and 4 had lung diseases (Table 2). None of them were pregnant, breastfeeding, had primary and secondary immuno-deficiencies, or diseases of the hematopoietic system. Thirty-two (32) of the 36 individuals with cardiovascular diseases had hypertension; one participant had myocarditis, one had pericarditis, and two of them had heart valve diseases (Table 2). When analyzing the influence of age on the observed parameters, the group was divided into groups with ten-year intervals (20–30, *n* = 4; 30–40, *n* = 25; 40–50, *n* = 34; 50–60, *n* = 26, and 60+, *n* = 14).

### 4.3. SARS-CoV-2 Serological Analyses

In this study, we used 5 different commercial SARS-CoV-2 ELISA tests to detect the humoral immune response against SARS-CoV-2. Details about the ELISA test used in this study are shown in Appendix A.

### 4.4. T-Cell Response

The SARS-CoV-2 specific T-cell response was determined by a commercial interferon-gamma (IFN γ) release assay (IGRA) using the Quant-T-Cell SARS-CoV-2 (product No. ET 2606-3003) and Quant-T-Cell ELISA (product No. EQ 6841-9601), manufactured by EUROIMMUN AG, Lübeck, Germany. The specific T-cell response was quantified according to the manufacturer’s instructions, with values > 100 mIU/mL marked as low positive, >200 mIU/mL marked as positive, and 100–200 mIU/mL as the grey zone.

### 4.5. Statistical Analysis

Statistical analyses were performed according to the protocols described by Hinkle et al. [41]. Groups were analyzed using analyses of variance ANOVA, post hoc comparisons were made by Tukey’s HSD *t*-test, and *p* < 0.05 was considered significant. Pearson’s correlation protocol was performed for correlation analysis.

### 4.6. SARS-CoV-2 Serological Analyses

SARS-CoV-2 serology was determined with semi-quantitative and quantitative commercial ELISA tests, as listed in Table A1. All tests were performed according to the manufacturer’s instructions, using a fully automated ELISA apparatus: EuroImmun I Analyzer (for EuroImmun tests) (Germany, Lübeck) and DS2 Dynex Technologies (for TestLine tests and Shanghai GeneoDx Biotech Co., Ltd.; Shanghai, China).

## 5. Conclusions

In our study, the effects of three doses of the Sinopharm vaccine and SARS-CoV-2 infection on the immune response against SARS-CoV-2 were investigated in 103 volunteers. Our results show that cardiovascular diseases increased the level of anti-N-IgG antibodies, while endocrinological diseases decreased neutralizing antibody and anti-N IgG antibodies, suggesting that these diseases may alter vaccine-induced immunity. A significant decrease in anti-S1 IgG levels was observed six months post-infection, and in anti-N IgG levels 18 months post-infection, while neutralizing antibodies and INF γ levels remained constant over the same periods. The study confirmed the emergence of “hybrid immunity”, a combination of natural and vaccine-induced immunity, and appears stronger and more durable than either form of immunity alone. A significant positive correlation was found between humoral and cellular immunity markers, suggesting a coordinated, COVID-19-specific response.

## Figures and Tables

**Figure 1 pharmaceuticals-17-00122-f001:**
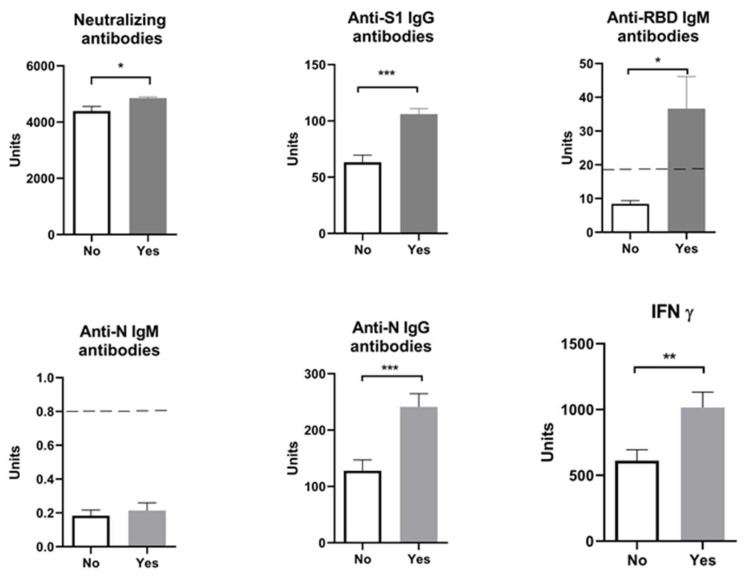
Markers of humoral and cellular immune response after vaccination with 3 doses of Sinopharm vaccine based on COVID-19 history. No—subjects had no COVID-19; Yes—subjects had COVID-19; * *p* < 0.05; ** *p* < 0.01 and *** *p* < 0.005; below ------- negative values.

**Figure 2 pharmaceuticals-17-00122-f002:**
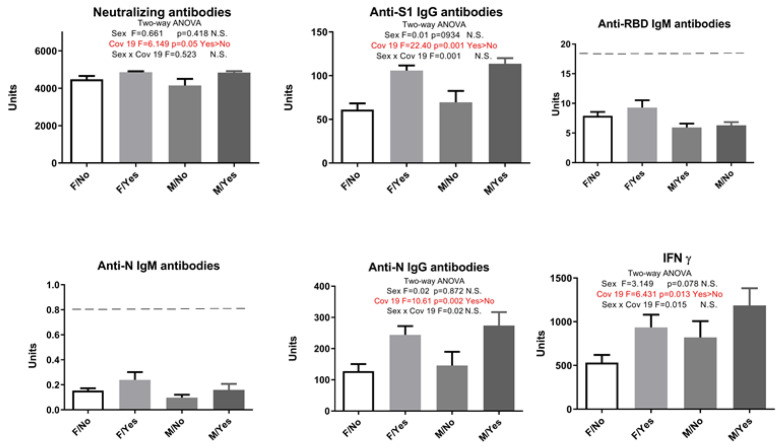
Markers of humoral and cellular immune response after vaccination with 3 doses of Sinopharm vaccine depending on sex and COVID-19 history (Cov19). F/No—women who did not have COVID-19; F/Yes—women who had COVID-19; M/No—men who did not have COVID-19; M/Yes—men who had COVID-19. N.S. Not significant; below ------- negative values.

**Figure 3 pharmaceuticals-17-00122-f003:**
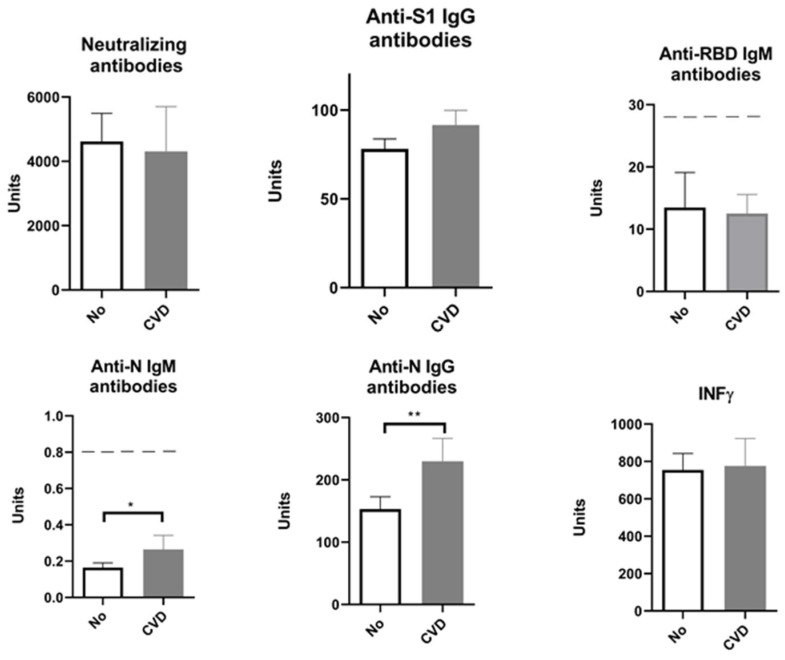
Markers of humoral and cellular immune response after vaccination with 3 doses of Sinopharm vaccine relating to the presence of cardiovascular diseases. No—participants who do not have cardiovascular disease; CVD—participants with cardiovascular diseases; below ------- negative values. * *p* < 0.05 and ** *p* < 0.01.

**Figure 4 pharmaceuticals-17-00122-f004:**
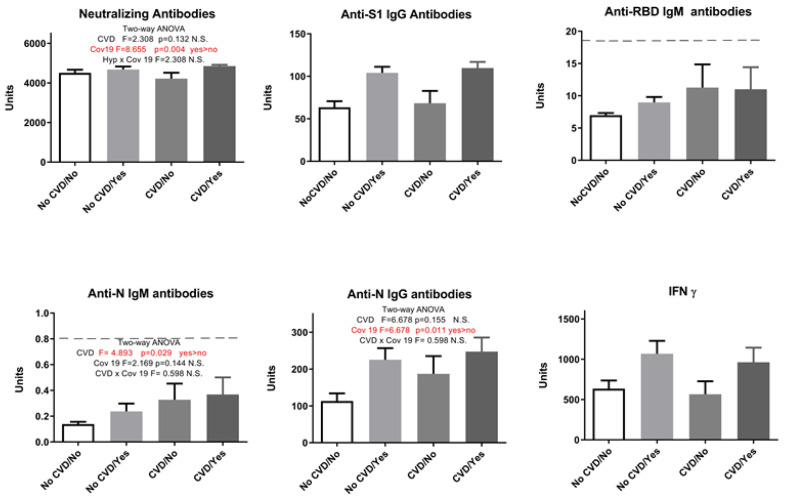
Markers of humoral and cellular immune response after vaccination with 3 doses of Sinopharm vaccine depending on the presence of cardiovascular diseases (CVDs) and COVID-19 history (Cov19). No CVD/No—participants who do not have cardiovascular diseases and who have not had COVID-19; No CVD/Yes—participants who do not have cardiovascular diseases and who have had COVID-19; CVD/No—participants with cardiovascular diseases who have not had COVID-19; and CVD/Yes—participants with cardiovascular diseases who have had COVID-19; below ------- negative values.

**Figure 5 pharmaceuticals-17-00122-f005:**
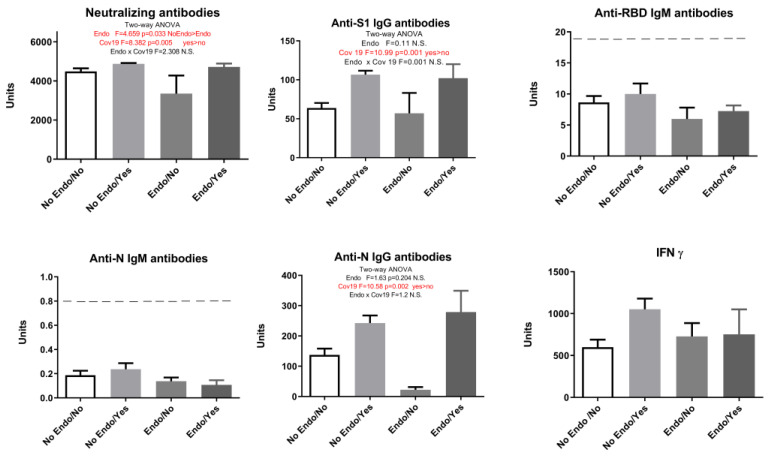
Markers of humoral and cellular immune response after vaccination with 3 doses of Sinopharm vaccine based on the presence of endocrine disorders (Endo) and COVID-19 history (Cov19). No Endo/No—subjects who do not have endocrine disorders and have not had COVID-19; No Endo/Yes—subjects who do not have endocrine disorders and have had COVID-19; Endo/No—participants who have endocrine disorders and who have not yet had a COVID-19; and Endo/Yes—subjects who have endocrine disorders and have had COVID-19; below ------- negative values.

**Figure 6 pharmaceuticals-17-00122-f006:**
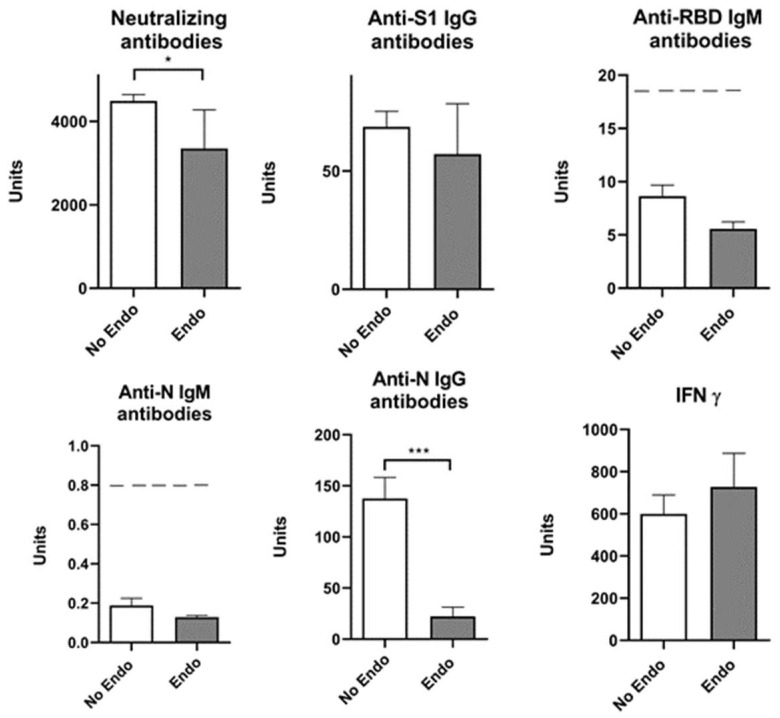
Markers of humoral and cellular immune response after vaccination with 3 doses of Sinopharm vaccine in the presence of endocrine disorders (Endo) and no COVID-19 history (No). No Endo/No—participants who do not have endocrine disorders and who have not had COVID-19; Endo/No—participants who have endocrine disorders and who have not had COVID-19; * *p* < 0.05 and *** *p* < 0.005; below ------- negative values.

**Figure 7 pharmaceuticals-17-00122-f007:**
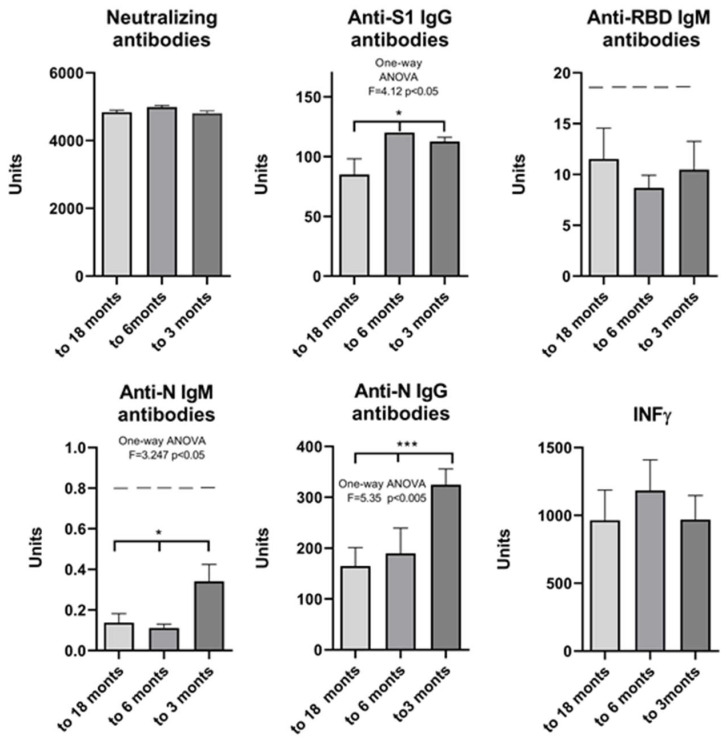
Markers of humoral and cellular immune response after vaccination with 3 doses of Sinopharm vaccine based on the time elapsed between the last clinical presentation of COVID-19 symptoms and the testing time. * *p* < 0.05 and *** *p* < 0.005; below ------ negative values.

**Figure 8 pharmaceuticals-17-00122-f008:**
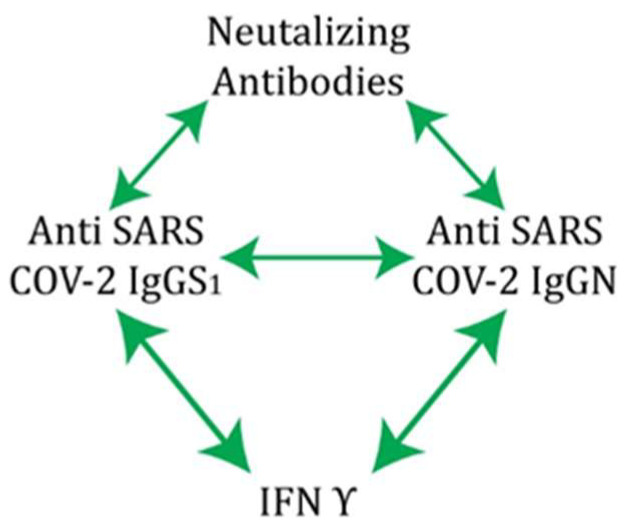
Schematic representation of the coordinated course of the immune response to vaccination.

**Table 1 pharmaceuticals-17-00122-t001:** Correlation analysis of neutralizing antibodies, IFN γ, anti-N IgG antibodies (Nucleocapsid), anti-RBD IgM antibodies (Receptor Binding Domen), anti-N IgM antibodies, and anti-S1 IgG antibodies (i RBD) performed on the complete sample. Red-colored numbers represent a statistically significant correlation between the examined parameters.

	Neutralizing Antibodies	Anti-S1 IgG Antibodies	Anti-RBD IgM Antibodies	Anti-N IgM Antibodies	Anti-N IgG Antibodies	IFN γ
Neutralizing antibodies						
anti-S1 IgG antibodies	0.400 ***					
anti-RBD IgM antibodies	0.0525	0.0923				
anti-N IgM antibodies	0.078	0.0002	0.098			
anti-N IgG antibodies	0.240 ***	0.666 ***	−0.099	−0.039		
IFN γ	0.142	0.392 ***	0.196	0.0923	0.336 ***	

*** *p* < 0.05.

**Table 2 pharmaceuticals-17-00122-t002:** Principal features of the participants included in the study.

Anamnestic Data					
Number of Participants					
Total number	103				
COVID-19 history	Yes	43	Before 3rd dose	16	
			After 3rd dose	27	
	No	60			
Sex	F	75			
	M	28			
Presence of cardiovascular diseases	Yes	36			Hypertension (*n* = 32)
					Myocarditis (*n* = 1)
					Pericarditis (*n* = 1)
					Heart valve diseases (*n* = 2)
	No	67			
Presence of diseases of the nervous system	Yes	1			
	No	102			
Presence of endocrinological diseases	Yes	11			Diabetes mellitus (*n* = 3)
					Thyroid gland diseases (*n* = 7)
					Pituitary gland diseases (*n* = 1)
	No	92			
Presence of liver diseases	Yes	2			
	No	101			
Presence of kidney diseases	Yes	1			
	No	102			
Presence of pulmonary diseases	Yes	4			
	No	99			
Presence of allergic reactions	Yes	13			
	No	99			
Presence of autoimmune diseases	Yes	2			
	No	101			

## Data Availability

Data are contained within the article.

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
