# Peer review of "Humoral and Cellular Immune Response after Three Doses of Sinopharm [Vero Cell]-Inactivated COVID-19 Vaccine in Combination with SARS-CoV-2 Infection Leads to Hybrid Immunity"

_pharmaceuticals, 2024, doi:10.3390/ph17010122_

Round 1
Reviewer 1 Report (Previous Reviewer 2)
Comments and Suggestions for Authors
It seems like the authors of the revised manuscript have addressed major points raised by me. They've provided additional details in their methodology, such as the statistical tests used and the comorbidities considered. For language and grammar concerns, they have performed a grammar check. They've clarified the definition of hybrid immunity and provided batch numbers for the vaccines used. The authors have added sections explaining which SARS-CoV-2 strains were in circulation during the study and included information about COVID-19 exposure history as requested. They've also made corrections to their figures and tables to reflect accurate data representation and ensure that comparisons are made appropriately within the context of COVID-19 exposure.
In light of these revisions, I recommend that the manuscript be accepted for publication. The study's findings on hybrid immunity and the implications for individuals with comorbidities are particularly noteworthy and will be of interest to the journal's readership.
Comments on the Quality of English LanguageMinor editing of English language required.
Author Response
Thanks for the kind words!We hope that the editing of English language is good enough.
Reviewer 2 Report (New Reviewer)
Comments and Suggestions for Authors
In the manuscript entitled ‘Hybrid immunity after a natural infection and vaccination with Sinopharm [Vero Cell] – Inactivated COVID-19’, the authors investigated the level of neutralizing antibodies, anti-S1 IgG, anti-RBD IgM, anti-N IgM, to anti-N IgG and interferon γ in vaccinated subjects with or without previous infection with SARS-CoV-2. Moreover, the authors divided the volunteers based on cardiovascular or endocrinological diseases trying to highlight differences among the immune response directed against SARS-CoV-2.
In my opinion, it could be an interesting topic, but the manuscript needs major revisions.
Major concerns:
- The title should be changed to better mirror the ‘take-home message’ of the authors. The current title does not explain what the authors want to demonstrate.
- In lines 55-56, the authors reported that neutralizing antibodies are anti-RBD and anti-S1 antibodies. However, in the article, the authors reported the distinct measurements of neutralizing antibodies, and of anti-S1 IgG and anti-RBD IgM. Could the authors explain what they mean as neutralizing antibodies?
- Could the authors specify the number of subjects in each age group?
- Lines 222-223, please add the reference for this affirmation or explain better.
- Line 264, please add the reference.
- The discussion section should be rewritten. I found many interesting insights; however they are not fully investigated; for example, it might be worth delving into further the difference between induced immune response between mRNA vaccine and Sinopharm vaccine.
- Could the authors hypothesize why in patients suffering from CVD or endocrinological diseases the immune response is different compared to the controls?
Minor concerns:
- There are some typos and grammatical errors.
- The quality of the graphs should be improved (in some graphs, the measuring unit is not shown)
Round 2
Reviewer 2 Report (New Reviewer)
Comments and Suggestions for Authors
This new version showed an improved quality.
The authors replied to all my comments.
This manuscript is a resubmission of an earlier submission. The following is a list of the peer review reports and author responses from that submission.
Round 1
Reviewer 1 Report
Comments and Suggestions for Authors
This scientific study investigates the humoral and cellular immune responses in individuals vaccinated with three doses of the Sinopharm COVID-19 vaccine, considering factors such as prior COVID-19 history, sex, cardiovascular diseases, endocrine disorders, and time elapsed since the last clinical presentation of COVID-19 symptoms.The study concludes that vaccination with three doses of the Sinopharm vaccine induces effective humoral and cellular immune responses, with considerations for prior COVID-19 history and underlying health conditions. The emergence of hybrid immunity is suggested to provide strong and sustained protection against COVID-19.
The sections are well built and the results are interesting, as well as provided discussion. The manuscript is ready to publish as it is.
Comments on the Quality of English LanguageEnglish language and style are fine/minor spell check required.
Reviewer 2 Report
Comments and Suggestions for Authors
- Major comments:
The study investigated the effect of three doses of the Sinopharm vaccine and SARS-CoV-2 infection on the immune response in 103 volunteers. It was found that cardiovascular diseases increased the level of anti-N-IgG antibodies, while endocrine diseases decreased the level of neutralizing antibodies and anti-N-IgG antibodies, indicating that these diseases can alter vaccine-induced immunity. A significant decrease in anti-S1 IgG levels was noted six months post-infection, and in anti-N IgG levels 18 months post-infection, while neutralizing antibody and interferon-γ levels remained constant over the same periods. The study confirmed the emergence of "hybrid immunity," which is a combination of natural and vaccine-induced immunity, and it seems to be stronger and more durable than either form of immunity alone. There was a significant positive correlation found between humoral and cellular immunity markers, suggesting a coordinated response specific to COVID-19.
The study contributes to understanding the long-term effects of vaccination and previous infection on the immune response to SARS-CoV-2, especially considering the presence of comorbidities like cardiovascular and endocrine diseases.
The main concerns are:
The manuscript does not sufficiently distinguish the findings from the current body of literature on COVID-19 immunity. The methodology section lacks the detail needed for replication, and the data presentation does not robustly support the conclusions drawn. The impact of comorbidities on vaccine response is a valuable aspect of the research; however, the analysis needs to be more rigorous to draw meaningful conclusions. Furthermore, the manuscript requires substantial revisions for language and grammar. The presence of numerous grammatical errors throughout the paper detracts from its readability and undermines the presentation of the research. I recommend thorough proofreading and editing by a native English speaker or professional language service to bring the manuscript to the publication standard.
- General concept comments
Here are some more considerations/suggestions for the study:
- Line 23, please provide a more precise definition of hybrid immunity in the introduction section.
- Line 65: since the samples were collected in 2022, the information on the Sinopharm vaccine should be detailed, like the catalog or batch number. Were these vaccines made from the inactivated wild-type SARS-CoV-2? Did the Sinopharm vaccine update after Omicron waves?
- Table 1. Please add rows to describe the COVID-19 exposure history, and whether the infection occurred before the vaccination or after.
- Line 141, The values of anti-RBD IgM antibodies in individuals who were vaccinated following a COVID-19 infection are statistically significant and enter the positive values (>18 U/mL). However, Figure 1 titled, Humoral, and cellular immune response after vaccination with 3 doses of Sinopharm vaccine based on previous COVID-19 history. The question is when infected with COVID-19, before or after the vaccinations?
- Figures 1-7, the Anti-N IgM antibodies detections were all below the detection limit, which may mean that the results of Anti-N IgM antibodies are not reliable. I suggest moving the result to supplementary.
- Figure 7: I have some concerns about Figure 7: were all the patients got COVID-19 infected before the vaccination? According to the text, some volunteers were not infected by COVID-19, while others did. Therefore, the comparisons here in the Figure should be done separately, like with COVID-19 exposure or not. The same problems apply to the lines of 344-346.
- Specific comments:
a. Line 17: why not use an anti-S1 IgM kit as an anti-RBD IgM kit may miss the NTD domain signal?
b. In lines 37-39, there were some problems regarding grammar.
c. Line 61-62, the information here is not true, especially for the updated mRNA vaccine.
d. Line 101, here it should be the Appendix B.
e. Line 140, here it should be the Figure 1.
f. Line 188, overcome? Or infection?
g. Line 197, here it should be “without” rather than “with.”
h. Figure 6, “/No” in the Figure here is redundant.
i. Line 298, not 103 subjects?

Poor.